# A Silk Fibroin Based Hydration Accelerator for Root Canal Filling Materials

**DOI:** 10.3390/polym12040994

**Published:** 2020-04-24

**Authors:** Ching-Shuan Huang, Sung-Chih Hsieh, Nai-Chia Teng, Wei-Fang Lee, Poonam Negi, Wendimi Fatimata Belem, Hsuan-Chen Wu, Jen-Chang Yang

**Affiliations:** 1School of Dentistry, College of Oral Medicine, Taipei Medical University, Taipei 110-31, Taiwan; jollyhuangtw12@gmail.com (C.-S.H.); endo@tmu.edu.tw (S.-C.H.); tengnaichia@hotmail.com (N.-C.T.); 2Department of Dentistry, Taipei Municipal Wan-Fang Hospital, Taipei 116-96, Taiwan; 3Department of Dentistry, Taipei Medical University Hospital, Taipei 110-31, Taiwan; 4School of Dental Technology, College of Oral Medicine, Taipei Medical University, Taipei 110, Taiwan; weiwei@tmu.edu.tw; 5School of Pharmaceutical Sciences, Shoolini University of Biotechnology and Management Sciences, Solan 173 212, India; poonam.546@shooliniuniversity.com; 6International Ph.D. Program in Biomedical Engineering, College of Biomedical Engineering, Taipei Medical University, Taipei 110-31, Taiwan; belemfatimataw@gmail.com; 7Department of Biochemical Science and Technology, National Taiwan University, Taipei 106, Taiwan; hcwu7@ntu.edu.tw; 8Graduate Institute of Nanomedicine and Medical Engineering, College of Biomedical Engineering, Taipei Medical University, Taipei 110-31, Taiwan; 9Research Center of Biomedical Device, Taipei Medical University, Taipei 110-52, Taiwan; 10Research Center of Digital Oral Science and Technology, Taipei Medical University, Taipei 110-52, Taiwan

**Keywords:** mineral trioxide aggregate, silk fibroin, SavDen^®^ MTA, ProRoot^®^ MTA

## Abstract

Mineral trioxide aggregate (MTA) is widely used in various dental endodontic applications such as root-end filling, furcal perforation repair, and vital pulp therapy. In spite of many attempts to improve handling properties and reduce the discoloration of MTA, the ideal root canal filling material has yet to be fully developed. The objective of this study was to investigate the setting time, mechanical properties, and biocompatibility of MTA set by a silk fibroin solution. A 5 wt% silk fibroin (SF) solution (a novel hydration accelerant) was used to set SavDen^®^ MTA and ProRoot^®^ white MTA (WMTA). Changes in setting time, diametral tensile strength (DTS), material crystallization, in vitro cell viability, and cell morphology were assessed by Vicat needle measurement, a universal testing machine, scanning electron microscopy (SEM), and WST-1 assay, respectively. The initial setting time of ProRoot^®^ MTA and SavDen^®^ MTA experienced a drastic decrease of 83.9% and 42.1% when deionized water was replaced by 5 wt% SF solution as the liquid phase. The DTS of SavDen^®^ MTA showed a significant increase after set by the SF solution in 24 h. A human osteoblast-like cell (MG-63)-based WST-1 assay revealed that both ProRoot^®^ MTA and SavDen^®^ MTA hydrated using SF solution did not significantly differ (*p* > 0.05) in cell viability. MG-63 cells with pseudopodia attachments and nuclear protrusions represent a healthier and more adherent status on the surface of MTA when set with SF solution. The results suggest that the 5 wt% SF solution may be used as an alternative hydration accelerant for MTA in endodontic applications.

## 1. Introduction

Mineral trioxide aggregate (MTA), a mixture of refined Portland cement and radiopacifier, is formulated as dental restorative materials for pulp capping, apexification, and root-end filling [1,2,3,4]. Mixing MTA powders with deionized water creates a hydration reaction to form calcium hydroxide (Ca(OH)_2_) and calcium silicate hydrate (CSH) providing unique antibacterial properties [5], sealing ability [6], biocompatibility [7], and promotion of hard-tissue formation [8,9]. MTA has been reported to have superior clinical efficacy and is less cytotoxic than other materials currently used in pulpal therapy [10]. However, MTA is difficult to use because of its long setting time [11], poor handling properties [12], and staining potential [13].

Currently, calcium lactate gluconate (CLG), a dual-function hydration accelerator, successfully improves the acquiescence of MTA by enhancing its clinical manageability and biocompatibility without aesthetic concerns caused by discoloration, which makes it more convenient for dentists during endodontic treatments [14,15]. Even though this modified MTA overcomes many previous drawbacks, it has been reported that the addition of CLG adversely affected its diametral tensile strength (DTS) set after one day [15].

Silk fibroin (SF), a protein harvested from *Bombyx mori* silkworm cocoons comprising a 26-kDa light chain and a 391-kDa heavy chain linked by a single disulfide bond forming an H–L complex [16], is a promising biomaterial for many biomedical applications such as drug delivery, wound healing, and bone tissue engineering [17,18,19]. Mieszawska et al. reported that SF combined with silica-based material showed an osteoinductive property for bone regeneration [20]. Inspired by the benefits of osteoinduction for SF/silica composites, SF with characteristics of biocompatibility makes it a suitable additive for tricalcium silicate-based MTA as root-end-filling materials [21,22]. The dominance of the β-pleated sheet crystals within the fibroin structure imparts the protein-based materials with superior mechanical properties but poor solubility [23]. There are many appropriate solvent systems for dissolving the silk fibroin. However, it is desirable to choose a solvent system with a favorable function for enhancing the hydration of MTA. Here, the calcium chloride could not only serve as a hydration accelerator for MTA [24], but also as a major ingredient in Ajisawa’s solvent system (CaCl_2_:EOH:water at 1:2:8 by mole) for SF [16,25,26]. Therefore, a novel SF-based hydration system for MTA was proposed.

In this study, our purpose was to identify if a 5 wt% SF solution would improve the handling and mechanical properties for MTA root canal filling materials.

## 2. Materials and Methods

To prepare the silk fibroin solution, we collected dried silkworm cocoons (Maioli District Agricultural Research and Extension Station, Council of Agriculture, Miaoli County, Taiwan) and degummed them with boiling aqueous 0.02 M Na_2_CO_3_ for 60 min. After degumming treatments, the extracted silk fibroin was washed in water repeatedly and dried in a 60 °C incubator. A three-phase solvent system comprising calcium chloride-ethanol-water in a molar ratio of 1:2:8 was used to dissolve the silk fibroin. The silk fibroin was completely dissolved at 60 °C for 24 h. As the extracted silk fibroin was being used as the solute, we mixed 5 g of silk fibroin with 95 g of the three-phase solvent to compose a 5 wt% SF solution. Finally, the 5 wt% SF solution was centrifuged at 6000 rpm for 5 min to eliminate impurity. This 5 wt% SF solution in liquid form was used to mix with MTAs for the following experiments.

### 2.1. Time to Initial Set

The liquid-to-powder (L/P) ratios were 1:4 and 1:3 for SavDen^®^ MTA (tYDS Biotech Inc., Taiwan) and ProRoot^®^ MTA (Dentsply Tulsa Dental, Tulsa, OK, USA), respectively. The initial setting times for MTAs set with deionized water (DDW) or 5 wt% SF solution were tested with a Vicat needle (Jin-Ching-Her, Yunlin County, Taiwan) (a movable rod weighing 300 g with a 1-mm-diameter needle) at an interval of 2 min. Initial setting times were recorded when the needle formed a compression mark less than 1 mm in depth in three separate areas.

### 2.2. Diametral Tensile Strength

The hydrated cements were placed into a cylindrical Teflon mold (5-mm height by 6 mm in diameter) and stored in a 100% relative humidity and 37 °C incubator for 24 h. The setting reaction was examined through the diametral tensile strength (DTS) test, which was assessed by a universal testing machine (CY-6040A8, Chun Yen Testing Machine, Taichung, Taiwan) at a cross-head speed of 0.5 mm/min. The DTS value of the cylindrical models was established from the equation of DTS=2P/πbw (from the axis of the cylinder), with *P* standing for the peak load (N), *b* for the diameter (mm), and *w* for the length (mm) of the cylinder.

### 2.3. Scanning Electronic Microscopy (SEM) of Cement Surfaces

The crystallization of hydrated cements on the cross-section areas was examined by SEM. The cement specimens were dehydrated and placed to cover glasses. Four cover glasses containing different samples were sputtered with gold palladium, and images were taken with a SEM (TM3030, Hitachi Ltd., Tokyo, Japan) in different magnifications.

### 2.4. Cell Culture, Cytotoxicity, and Proliferation Assay

The cell viability of the extracted mediums of the four cement sets was evaluated with MG-63 osteoblast-like cells. The cells were expanded in Dulbecco’s modified Eagle’s medium (DMEM, HyClone, Logan, UT, USA) supplemented with l-glutamine (4 mmol/L), 10% fetal bovine serum (FBS), and 1% penicillin-streptomycin at 37 °C and 5% CO_2_. The extracts for the test were prepared by immersing specimens in medium without FBS at an extraction ratio of 0.2 g/mL for 37 °C for 48 h. Then, five times dilution was processed for the aqueous extracts and FBS was added to compose the extract medium for culturing. Cells of 5 × 10^4^/well in a 24-well microplate were incubated at 37 °C for 24 h. After 24 h, 500 μL of extract medium from the four groups, positive control (1% dimethyl sulfoxide (DMSO)), negative control (high-density polyethylene), and control medium (fresh medium) were exchanged as the culturing medium for the monolayer cells. After incubation at 37 °C for 24 h, cell proliferation rate was then evaluated (Cell Proliferation Reagent WST-1, Roche, Mannheim, Germany) [27], and culture media were collected for quantification of the lactate dehydrogenase (LDH) release (CytoTox 96^®^ Non-Radioactive Cytotoxicity Assay, G1780; Promega, WI, USA).

### 2.5. Scanning Electronic Microscopy (SEM) of Cell Morphology

MG-63 osteoblast-like cell morphology on cement surfaces was examined by SEM. Cells at 2 × 10^5^/well were seeded on the four groups of set cement surfaces with 4 mL of complete medium for 5 days. Cell-containing specimens were fixed with 2.5% glutaraldehyde and 2% paraformaldehyde for 20 min and post-fixed with 1% osmium tetroxide for 1 h. Dehydration was carried out in an ethanol series at concentrations of 70%, 80%, 90%, 95%, and 100% in a critical point dryer (HCP-2, Hitachi Ltd., Tokyo, Japan). Finally, a thin layer of gold was sputtered on the specimens and images were taken with a scanning electron microscopy (SU3500, Hitachi Ltd., Tokyo, Japan).

### 2.6. Data Analysis

The one-way ANOVA test was used to evaluate the statistical significance of the measured results. When the analysis indicated significant variances between group means, each group was compared using Turkey’s multiple comparison test. Results of *p* < 0.05 were considered statistically significant.

## 3. Results

### 3.1. Initial Setting Time and Handling Properties

The initial setting times of each cement powder-liquid system are presented in Table 1. Replacing deionized water (DDW) with 5 wt% SF solution reduced the setting times for ProRoot^®^ MTA from 157.5 ± 8.9 to 25.3 ± 2.1 min and SavDen^®^ MTA from 17.5 ± 1.8 to 10.1 ± 0.9 min. Based on the CaCl_2_ component of the SF solution, it acts as an accelerator for MTA setting. The handling property of the white MTA and DDW mixture was grainy and sandy, making usage with the root-end cavity and compaction of pulp capping difficult. Unlike setting with DDW, the 5 wt% SF solution dramatically improved handling properties for both MTAs, especially in the mixture with SavDen^®^ MTA. The structure of SF created a denser and thicker texture for the cement, which is similar to the handling properties of typical IRM.

### 3.2. Mechanical Physical Properties

To evaluate the mechanical properties of dental composites, DTS is an acceptable common test. Figure 1 shows DTS values of SavDen^®^ MTA and ProRoot^®^ MTA mixed with DDW and the 5 wt% silk fibroin solution, respectively. The DTS for hydrated SavDen^®^ MTA by DDW and the 5 wt% silk fibroin solution were significantly improved from 1.15 ± 0.05 MPa to 2.82 ± 0.39 MPa at a 24-hour setting (*p* < 0.001). However, the DTS values of ProRoot^®^ MTA were statistically higher than SavDen^®^ MTA no matter what setting periods and hydration liquids were used. Unlike the ProRoot^®^ MTA set by 5 wt% SF solution, the DTS value of ProRoot^®^ MTA did not elevate when using SF solution.

### 3.3. Crystallization of Cement Surfaces

Studies on the microstructures of MTA pastes have given results broadly similar to those on Portland cement pastes due to its use at 70–80% in various MTAs. The particle size and crystallization of cements on the cross-section areas were examined by scanning electron microscopy (SEM). The microstructure of hydrated MTA consists of cubic, needle-like crystals, and porous structures. In Figure 2, the image shows that SavDen^®^ MTA set by the 5 wt% SF solution has more cubic and needle-like morphologies of 2–5 μm in length over the surface of the material, which provides the material an important interlocking of the entire mass. The particle size on the surface of ProRoot^®^ MTA is larger when set by DDW than that of the 5 wt% SF solution.

### 3.4. Viability of Osteoblast-like Cells on MTA

Cell cytotoxicity was evaluated by examining extracellular LDH released by lysed cells where higher optical density value represents higher cytotoxicity. In Figure 3a, the groups of MG-63 cells seeded on different hydrated cement/liquid systems showed no cytotoxic effect in the LDH assay for all experiment groups. Cell proliferation assay in Figure 3b showed an obvious result of the survival of MG-63 cells in all cement groups. In the WST-1 assay, the viability of osteoblast-like cells in SavDen^®^ MTA set by the SF solution (optical density value was 2.89 ± 0.29) was significantly higher than the negative control group (optical density value was 2.45 ± 0.19, *p* < 0.05). In addition, the cell viability of the groups added with the SF solution was slightly greater than that of deionized water hydrated groups.

### 3.5. Cell Morphology on Cement Surfaces

Figure 4 revealed MG-63 cells spreading randomly on the hydrated MTA surface under the scanning electron microscope. On the surface of MTA set by the SF solution, cells were observed with more nuclear protrusions and propagated cytoplasmic extensions, which helped cells to better adhere to the surface. Porous microstructures with acicular crystals were observed in the surface of all four cement groups—(a) ProRoot^®^ MTA + DDW, (b) ProRoot^®^ MTA + SF solution, (c) SavDen^®^ MTA + DDW, and (d) SavDen^®^ MTA + SF solution. Two types of hexagonal crystals were reflected in (c) and (d), which are planar-like structures (marked with arrow) and pillar-like structures (marked with hollow arrow). In addition, acicular sheet-like crystals (marked with star) forming in layers were contained in (d) [28].

## 4. Discussion

The major goal of regenerative endodontics is to treat immature teeth with pulp necrosis and underdeveloped roots for further potential root maturation and return of vitality [29]. A surgical procedure usually involves the placement of a medicament designed to seal off the root canal contents from the periradicular tissues and repair root defects [30]. Among endodontic repair materials, MTA was widely recognized as an effective root canal filling material [31]. MTA mainly comprises of tricalcium silicate (C3S), dicalcium silicate (C2S), radiopacifier, and gypsum [32]. In cement chemistry, hydration denotes the changes when an anhydrous cement is mixed with water. However, this chemical reaction is more complex than simple conversions of anhydrous compounds into the corresponding hydrates [33]. The hydration reaction of MTA was reported to be similar to that of Portland cement shown as Equation (1) and Equation (2) [34].



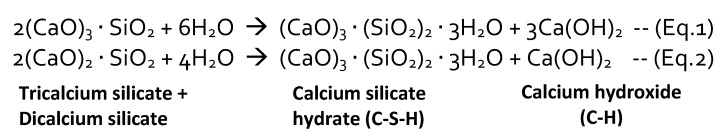



The setting reaction of MTA and Portland cement is a complex process. From the phenomenological standpoint, hydration starts when the calcium, silicon, and hydroxide ions are released from the surface of the calcium silicates [35]. Once the accumulating concentration over the solubility limit of calcium hydroxide (CH, Ca(OH)_2_) or calcium silicate hydrate (C-S-H) precipitation occurs, it results in a colloidal gel that solidifies to a hard structure. C–S–H is a nanoporous and nonstoichiometric substance with the calcium-to-silicon molar ratio (Ca/Si) range of 0.8 to 2.0. Thus, its composition is sensitive to the composition of the surrounding solution [36]. After hydration, the resulted byproducts of C-S-H offer great sealability; while the CH provides an antibacterial environment under a high alkaline pH of ~12.5 [37].

Certain treatments such as root perforation, pulp capping, and root-end filling are highly dependent on the setting time of the material [38]. The short setting time can prevent washout effects of excessive bleeding and promotes obturating a tooth within one visit [39]. To deal with the long setting time of MTA, the addition of a calcium compound is an effective approach. Ber et al. claimed an addition of 1% methylcellulose/calcium chloride (MC/CaCl_2_) into MTA resulted in an initial setting time drop from 202 to 57 min and improvement in its compressive strength and moldability [40]. Kogan et al. reported 5% CaCl_2_ decreased the setting time from 50 min (sterile water alone) to 20–25 min [41]. In this study, the function of SF and CaCl_2_ can serve as the viscosity enhancer and setting accelerator. In contrast to setting with DDW, 5 wt% SF solution can drastically improve the handling properties and reduce the setting times for ProRoot^®^ MTA from 157.5 ± 8.9 to 25.3 ± 2.1 min and SavDen^®^ MTA from 17.5 ± 1.8 to 10.1 ± 0.9 min. SavDen^®^ MTA with the inclusion of CLG shortened the setting time to less than 20 min and solved the aesthetic concerns [14]. A comparison of different accelerators of calcium chloride, calcium nitrite/nitrate, and calcium formate on GMTA, WMTA, and PC was investigated. It was found that all three accelerators significantly accelerated the setting time of GMTA and PC; only calcium chloride and calcium formate significantly accelerated WMTA [42]. Calcium chloride and low-dose citric acid used as a hydration accelerator could decrease the setting time of MTA. Hydration accelerators increase the osteogenic effect and show a similar effect on the mineralization of MTA, which may have clinical applications [43].

The amino acid composition of SF from *Bombyx mori* consists mainly of a nonpolar group of Gly (43%) and Ala (30%) forming the hydrophobic anti-parallel β-sheet crystallite domains with a repetitive hexapeptide sequence of Gly-Ala-Gly-Ala-Gly-Ser in a heavy chain [44]. The amino acid sequence of the L-chain is a non-repetitive forming amorphous region, so the L-chain is more hydrophilic and relatively elastic [45,46]. The excellent handling properties of the 5 wt% SF solution set MTA during hydration might be attributed to viscosity enhancement from the hydrophilic and amorphous characteristics of SF protein macromolecules. In addition, the β-sheet crystallite domains of SF might play a role in the mechanical properties of reinforced MTA in a solid state.

Although mechanical properties of MTA do not directly relate with their clinical efficacy, testing can provide valuable information about the consistency of product quality and formulations. Portland cement concrete is a relatively brittle material; thus the mechanical behavior of hydrated cement is critically influenced by crack propagation under fracture mechanics. In addition, many other factors such as formulation, L/P ratio, porosity, volume content, setting environment, setting time period, and reinforcement are also of great importance. However, the governing factors are not always in a linear relationship. For example, incorporation of 5 wt% alkali resistant (AR) glass fiber can significantly provide the reinforcement effect to ProRoot MTA, but the DTS showed a ~17% drop when increasing AR glass fiber to 10 wt% [47].

ProRoot^®^ MTA showed significantly higher DTS than that of SavDen^®^ MTA set by DDW at one day. However, SavDen^®^ MTA revealed significant improvement of DTS from 1.15 ± 0.05 MPa to 2.82 ± 0.39 MP compared with set by the 5 wt% SF solution to DDW at day one. The initial setting time might be dependent on how fast the concentration increases over the product of solubility (Ksp); however, the DTS is significantly affected by crystallization kinetics of CH and C-S-H formation. Our results suggested that a 5 wt% SF solution can reduce the time to reach the solubility limit of CH and C-S-H, but it retarded the crystallization of CH and C-S-H.

MTA is a type of mineral cement which solidifies as a hard structure in an interlocking mass upon hydration. The major phases of hydrated MTA are C-S-H and [Ca(OH)_2_], which are produced as by-products later during the hydration process. Lee et al. reported this crystallization consists of the formation of cubic and needle-like crystals evidenced by SEM after set for seven days [48]. Multilayered nanocrystalline structure were formed for hydrated calcium silicate (CS). In the early stage of hydration, the newly formed C-S-H crystals are acicular or fibrous shaped. As hydration progresses, these C-S-H crystals form the ground mass. Finally, the hexagonal plate-shaped and hexagonal column-shaped crystals on the surface of hydrated CS are most likely Ca(OH)_2_ [28]. However, the facetted crystals usually are attributed to CaCO_3_ due to the carbonation of Ca(OH)_2_ with CO_2_ in the atmosphere.

The origin of needle-like structures for hydrated MTA seems quite diverse. Jiménez-Sánchez et al. reported the microstructure of 50–100 nm needle-like morphologies over the entire surface of the material was the un-hydrated precursor for MTA HP Repair^®^ under SEM [49]. Kirchheim et al. claimed the needles should be ettringite or hydrated calcium aluminum sulfate hydroxide (Ca_6_Al_2_(SO_4_)_3_(OH)_12_·26H_2_O) formed by the reaction between sulfate ions and the aluminate phases in cement due to the fast reaction of orthorhombic tricalcium silicate with gypsum [50]. Further studies are needed to clarify the needle structure developed when SavDen^®^ MTA is hydrated with an SF solution.

The results of MG-63 cells seeded on different hydrated cement/liquid systems showed no significant difference in cytotoxic effect among all the experimental groups. In fact, silk fibroin is a kind of fibrous protein. Unlike globular proteins comprised tertiary or quaternary structures, fibrous proteins have mechanical and structural functions but no biological functions. The osteoinduction was observed in silk fibroin/silica composite with upregulation of gene expression in bone sialoprotein (BSP) and collagen type 1 (Col 1) osteogenic markers [20]. Furthermore, silk fibroin/hydroxyapatite composite hydrogel induced by gamma-ray irradiation showed increased cell proliferation, adhesion, and hMSCs osteogenic differentiation [51]. The potential advantages of osteoinduction or osteogenic differentiation from silk fibroin are desirable to implement to SF/MTA. Extensive in vivo studies are needed to verify this possibility.

## 5. Conclusions

According to the findings of this study, both SavDen^®^ MTA and ProRoot^®^ MTA set with silk fibroin solution showed a dramatic reduction in setting time and an improvement in handling properties. Easier manipulation, faster setting time, and stronger diametral tensile strength are the major advantages of using a 5 wt% SF solution with MTA. The enhanced properties of SF hydrated cements may be more effective. The expression of mineralization markers as well as the in vivo evaluation may need to be further investigated to verify their usefulness in clinical applications.

## Figures and Tables

**Figure 1 polymers-12-00994-f001:**
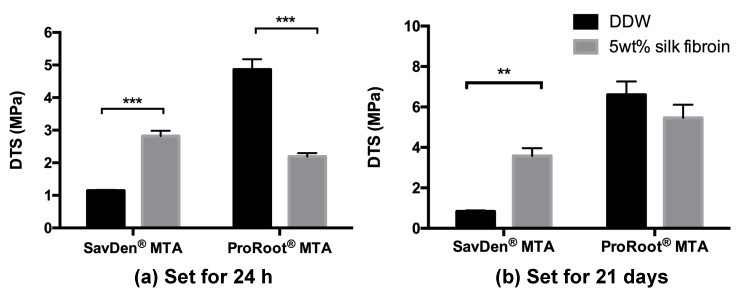
The diametral tensile strength (DTS) value of mineral trioxide aggregate (MTA) cements set by DDW and silk fibroin solution after (**a**) 24 h setting and (**b**) 21 days setting. ** Indicates a significant difference (*p* < 0.01). *** Indicates a significant difference (*p* < 0.001).

**Figure 2 polymers-12-00994-f002:**
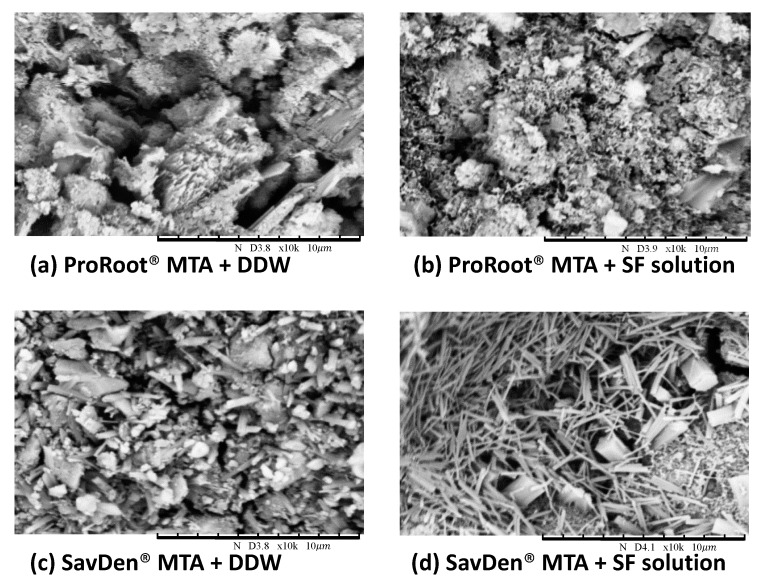
SEM micrographs of the cross-section area in various cement surfaces with the magnification of ×10,000. (**a**) ProRoot^®^ MTA + DDW has a larger particle size compared to those of (**b**) ProRoot^®^ MTA + SF solution. (**d**) SavDen^®^ MTA + SF solution contains additional amounts of cubic and needle-like structures, while (**c**) SavDen^®^ MTA + DDW has scattered particles compared to (**d**).

**Figure 3 polymers-12-00994-f003:**
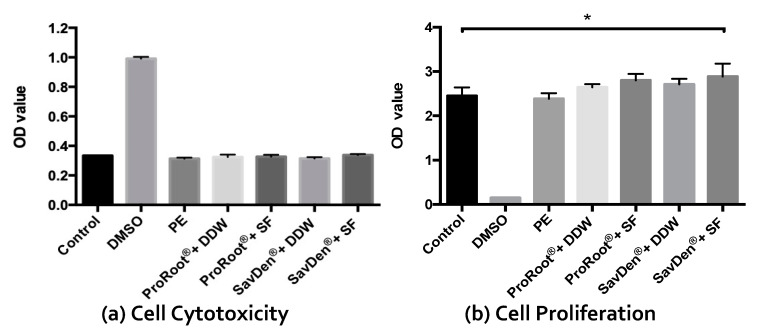
MG-63 osteoblast-like (**a**) cell cytotoxicity and (**b**) cell proliferation of ProRoot^®^ white MTA and SavDen^®^ MTA individually set with deionized water (DDW) and 5 wt% SF solution. Three samples (n = 3) were examined for each data point. * Indicates a significant difference (*p* < 0.05).

**Figure 4 polymers-12-00994-f004:**
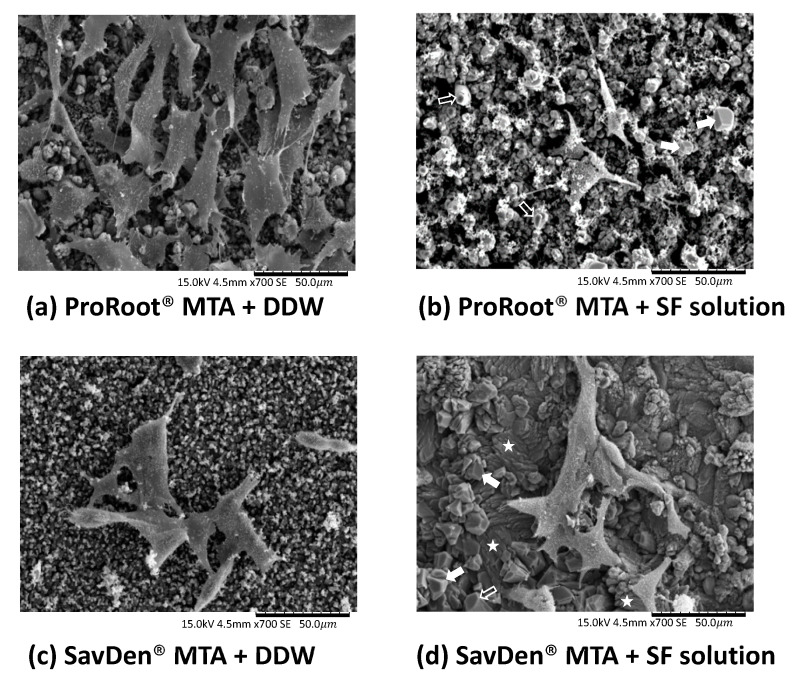
The SEM micrographs demonstrated the morphological features and attachment of MG-63 osteoblast-like cells on surfaces of various set MTAs after hydration for one day and seeded for five days in 37 °C (original magnification, ×700). (**a**) ProRoot^®^ MTA + DDW and (**c**) SavDen^®^ MTA + DDW are micrographs of MG-63 cells attaching to cement surfaces set by DDW. In (**b**) ProRoot^®^ MTA + SF solution and (**d**) SavDen^®^ MTA + SF solution, two types of hexagonal crystals were marked with arrow. Pillar-like structures were marked with hollow arrow. Hexagonal crystals with planar-like structures and pillar-like structures were marked with arrow and hollow arrow, respectively; while acicular sheet-like crystals were marked with star.

**Table 1 polymers-12-00994-t001:** Means of initial setting time as well as handling properties of different cement powder-liquid system.

Powder	SavDen^®^ MTA		ProRoot^®^ White MTA	
Liquid	5 wt% SF solution	water	5 wt% SF solution	water
Initial setting time (min)	10.1 ± 0.9	17.5 ± 1.8	25.3 ± 2.1	157.5 ± 8.9
Handling properties	Excellent	Good	Excellent	Very poor (sand feel)

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
