# Peer review of "A Silk Fibroin Based Hydration Accelerator for Root Canal Filling Materials"

_polymers, 2020, doi:10.3390/polym12040994_

Round 1
Reviewer 1 Report
Dear Authors
This idea of mixing SilkFibroin is very new and biocompatible. I would recommend all authors to do more experiment on this hybridization.
Few suggestions before the acceptance of this manuscript is required;
a) add these book chapters information in introduction part where applicable and cite them.
- https://www.sciencedirect.com/science/article/pii/B9780081024768000128
- https://www.sciencedirect.com/science/article/pii/B9780081024768000013
https://www.sciencedirect.com/science/article/pii/B9780081024768000025
Regarding the Discussion heading add the information form below paper;
https://www.sciencedirect.com/science/article/pii/S165836121400033X
The overall paper needs English editing from some native speakers.
Reviewer 2 Report
I have reviewed the manuscript “A Silk Fibroin Based Hydration Accelerator for Root Canal Filling Materials” submitted to “Polymers” for publication. In this manuscript, authors have added 5 wt% Silk Fibroin solution to the MTA root canal filling materials and investigated the mechanical properties, setting time, and biocompatibility. The manuscript fits well within the scope of the journal; however, this manuscript needs further improvements; there are some suggestion for authors.
The use of English language is reasonable, however, there are a number of punctuation and grammatical errors; that should be corrected and rephrased using academic English for a better flow of text for reader. Authors may consider language editing for this manuscript.
Line 38-39: is not clear, please rephrase it for further clarity
Line 50: please replace “filling materials” with “restorative materials”
Line 60-62: “Even though this modified MTA…..; please cite a reference for this?
Line 73: Ajisawa's solvent system… please cite original reference of Ajisawa's as well
Ajisawa, Akiyoshi. "Dissolution of silk fibroin with calciumchloride/ethanol aqueous solution." The Journal of Sericultural Science of Japan 67.2 (1998): 91-94.
In the introduction; information about the silk fibroin should be expanded; authors could include the following paper of Prof; Kaplan in the introduction and/or discussion sections
Functional material features of Bombyx mori silk light versus heavy chain proteins. Biomacromolecules. 2015 Feb 9;16(2):606-14.
Only using Ajisawa's solvent system does not justify the significance of this study; please elaborate on it?
Materials and Methods: The information regarding the silk fibroin process is vague; author should provide further details? Did authors use silk in Ajisawa, solvent? With highionic concentration of calcium chloride? Was that dialysed to get the aqueous solution? How and what format of 5% silk solution was used to mix with cement? How 5% was calculated etc
Figure 2: shows nice images; please add to captain the key features shown in SEM images; also add bold scale bars
Figure 4: description and scale bars are good
Line 247-255: this description of secondary conformation is based on natural silk; however same is not possible one dissolved in ionic solvent; the present study has not checked the conformation once in solution or mixed to cement?
Also include limitations of the study if any? Such as chemistry of the combination has not been included; high about of CaCl2 if coming from the solvent may need further investigation too.
Reviewer 3 Report
The authors conducted a study with the physical-mechanical and biocompatibility characteristics of an MTA cement with the addition of a silk fibroin solution. Only a few changes are to be considered and made.
- “Table 1. Means of initial setting time as well as handling properties of different cement powder-liqui…….” Move Above, as indicated by the arrow
see attached file
- Enter what are the limitations of the study
- “Even though this modified MTA overcomes many previous 61 drawbacks, it has been reported that the addition of CLG adversely affected its diametral tensile 62 strength (DTS) set after 1 day.” Insert Bibliography Reference
- Linea 254 Insert Bibliography Reference
- Linea 260 Insert Bibliography Reference
- “2.4 Cell Culture, Cytotoxicity, and Proliferation Assay” Why has cytotoxicity not been assessed with the MTT assay?

Author Response
Please attachment file!

Round 2
Reviewer 1 Report
Dear Author
Well improved.
Reviewer 2 Report
Many thanks for the revision and incorporating all suggested changes to the manuscript for which I am very thankful.
Reviewer 3 Report
The authors answered my doubts and made the requested changes